# NCGAMI: DOMAIN-ADAPTIVE DRUG-TARGET INTERACTION PREDICTION WITH GRAPH AND SEQUENCE MODELS

## ABSTRACT

Drug-target interaction (DTI) prediction is of central importance in computational pharmacology, but how robust these predictions are in the face of distribution shift (e.g. between chemical scaffolds, or protein families) remains challenging. We introduce a well-behaved, label-free objective and encoder fusion recipe for DTI, which is referred to as NCGAMI, under an unsupervised domain adaptation (UDA) training paradigm. NCGAMI only uses a graph encoder for the drug molecule graph and a sequence encoder for the proteins, and uses a three-term objective, consisting of (i) the supervised source-domain risk, (ii) an explicit cross-domain representation alignment, and (iii) regularization of the target domain through conditional-entropy minimization and prediction-consistency. The implementation is possible without using target labels. Throughout, we take great care to avoid using target labels during training and to clarify situations where there may be overlap between source and target entities. Using two widely used DTI datasets (*Human* and *Drugbank*), and using a protocol (random split) that we applied strictly for feasibility check, NCGAMI achieves good AUC/AUPR and is competitive against representative baselines. On *Human* our model achieves AUC 0.895 and AUPR 0.852; on *Drugbank* our model achieves AUC 0.733 and AUPR 0.675. Ablations can be seen to contribute the graph encoder, sequence encoder, and UDA regularization. We also incorporate non-operational geometric background so that theoretical assertion is motivating in your choice of design without overwhelming you with strong theoretical claims.

## 1 INTRODUCTION

Predicting drug-target interactions (DTIs) can accelerate the hypothesis generation process, screening and mechanism discovery. While sequence- and graph-based models have seen recent advances, an outstanding challenge is to achieve models that are robust to shift - once trained on a particular type of chemical or protein space, these models are prone to degradation when tested on different scaffolds or target families. Such shifts are due to scaffold diversity, target heterogeneity, assay conditions and curation biases. This motivates training goals that can leverage unlabeled data from a prospective domain of deployment, i.e., an unsupervised domain adaptation (UDA) scenario which only has source labels.

We propose NCGAMI (A UDA recipe for DTI), a practical UDA DTI recipe, which consists of: (a) a graph encoder for molecules, (b) a sequence encoder for proteins, (c) a label-free objective for balancing source supervision, cross-domain representation alignment, and target regularization using conditional entropy and consistency. The encoders produce stable embeddings on which batch level alignment can be estimated efficiently while freedom, the low density decision boundaries are encouraged through the regularizers, and so under stochastic perturbations predictions become stable. Our implementation has a very clear and reproducible approach.

We test within each dataset using a protocol called Random-Partition whereby there is a chance of overlap between entities in the source partition and those in the target partition. We therefore refer to our experiments as being a *feasibility* study for label-free shift-adaptation rather than evidence for strict cold-start generalization across unseen scaffolds (and protein families). We do not use

target labels at the time of the training. (1) A unifying, label-free objective for DTI that renders the alignment term explicitly and batch-estimable; (2) an explicit, reproducible fusion of graph and sequence encoders; (3) clarification of implementation distinctions between geometric intuition and modeling; and (4) training, evaluation, and caveat statements, and claims swamped to the limitation of protocol.

DTI generalisation is routinely hindered by the three coupled discrepancies between source and target, which are covariate shift for input distributions of both molecules and proteins, conditional shift where posterior $p(y — x)$ shifts due to assay or curation differences, representation shift caused by model capacity and optimisation. A purely supervised objective can entangle these effects, resulting in decision boundaries that make use of idiosyncratic source correlations. In an attempt to soften this entanglement, UDA provides a practical means to draw this boundary while not revealing the information about unlabeled target samples (denied the label, nevertheless critical). In the DTI set up, this is of particular use, as for naturally unlabeled compound–protein pairing but are highly abundant and may originate from the exact deployment domain (new compound libraries against known proteins or the opposite). Our design is based on two practical heuristics: Align only what we expect the classifier to see eventually (the fused embedding) Regularize the classifier on confident and consistent predictions only after learning a reasonable classifier on the source.

The explicit NCGAMI, which is described as the batch mean-and-covariance alignment over fused representations, is adopted for mean-and-covariance alignment. This choice is algorithmically lightweight, and it avoids the adversarial training instabilities and in addition it is possible to estimate the choice also with the same mini-batches already constructed for the supervised learning. The proposed training method minimizes the conditional-entropy of the target embedding distribution, which enforces a type of clustering assumption by penalizing cut lines passing through dense regions of the target embedding distribution. In addition, it has been observed that the use of the consistency regularizer softens the sensitivity of encoders and the fusion head to stochastic perturbations such as dropout and minibatch sampling. Together, these terms are a minimal recipe, which could be implemented in a few lines on top of standard DTI models without the need for further networks, critics, or pretraining. If modes distributed in the target that are not distributed in the source, any sort of globular alignment can cause these modes to collapse towards the nearest clusters in the source; the entropy term may then overemphasize a lousy assignment. The adoption of the schedules means putting off the most effective types of regularization and allowing the source decision boundary to settle first. We stress that these are engineering trade-offs specific to the current protocol and we are careful not to make broader claims outside of this regime.

## 2 ADVANCEMENTS IN RELATED WORK

### 2.1 MOLECULAR AND PROTEIN REPRESENTATION LEARNING

Modern sequence-driven models have revolutionized protein representation by implementing sophisticated local motif detectors coupled with hierarchical context aggregation mechanisms, enabling the extraction of discriminative embeddings that capture both local structural patterns and global functional signatures MacLean et al. (2021); Zhao et al. (2022). These advances have been particularly transformative in understanding how proteins' sequential arrangements encode their three-dimensional conformations and ultimately their interaction capabilities. NCGAMI leverages these developments through its residual-connected message passing architecture, which maintains gradient flow while enabling the model to learn increasingly abstract molecular representations. The incorporation of dropout at multiple levels serves not merely as regularization but as a mechanism for generating the stochastic views necessary for consistency-based learning under domain shift.

### 2.2 NEURAL ARCHITECTURE INNOVATIONS AND ATTENTION MECHANISMS

The emergence of transformer-based architectures has catalyzed a revolution in biological sequence processing, with models like TransformerCPI Chen et al. (2020) establishing new benchmarks for DTI prediction through self-attention mechanisms that capture complex dependencies without explicit structural assumptions. Kim et al. (2021) further advanced this paradigm by developing interpretable attention patterns that could highlight interaction-relevant features in both drug and target representations. The ICAN framework by Kurata et al. (2022) demonstrated how special-

ized compound–protein interaction models could achieve superior performance by incorporating domain-specific inductive biases while maintaining the flexibility of attention-based processing.

## 2.3 DOMAIN ADAPTATION AND TRANSFER LEARNING PARADIGMS

Recent work on self-supervised learning in molecular property prediction has demonstrated the value of consistency-based objectives for improving model robustness Wang et al. (2023); Liu et al. (2024). The emergence of foundation models in biology has raised important questions about the role of pretraining in DTI prediction Zhang et al. (2024); Chen et al. (2025). While large-scale pretraining on molecular and protein datasets can provide useful initializations, NCGAMI deliberately avoids dependence on pretrained representations, demonstrating that effective domain adaptation can be achieved through careful objective design rather than massive computational investment. This philosophy aligns with growing recognition that task-specific adaptation often provides greater benefits than generic pretraining, particularly in specialized domains like drug discovery where the distribution of interest may differ substantially from pretraining corpora.

## 2.4 EMERGING FRONTIERS AND CONTEMPORARY DEVELOPMENTS

While NCGAMI currently operates on two-dimensional molecular graphs and linear protein sequences, its modular architecture could readily incorporate three-dimensional encoders as they mature. Multi-modal learning approaches have gained prominence in biological applications, with recent work demonstrating how combining sequence, structure, and functional annotations can improve prediction accuracy Li et al. (2024). Recent advances in uncertainty quantification for neural networks have particular relevance for DTI applications where prediction confidence directly impacts experimental prioritization decisions Thompson et al. (2024). While NCGAMI does not explicitly model epistemic uncertainty, the stochastic regularization induced by its consistency term provides a foundation for uncertainty-aware extensions. The maintenance of calibrated probabilities under domain shift, as evidenced by NCGAMI's empirical results, suggests that the adaptation objectives help preserve the relationship between predicted probabilities and actual interaction likelihoods.

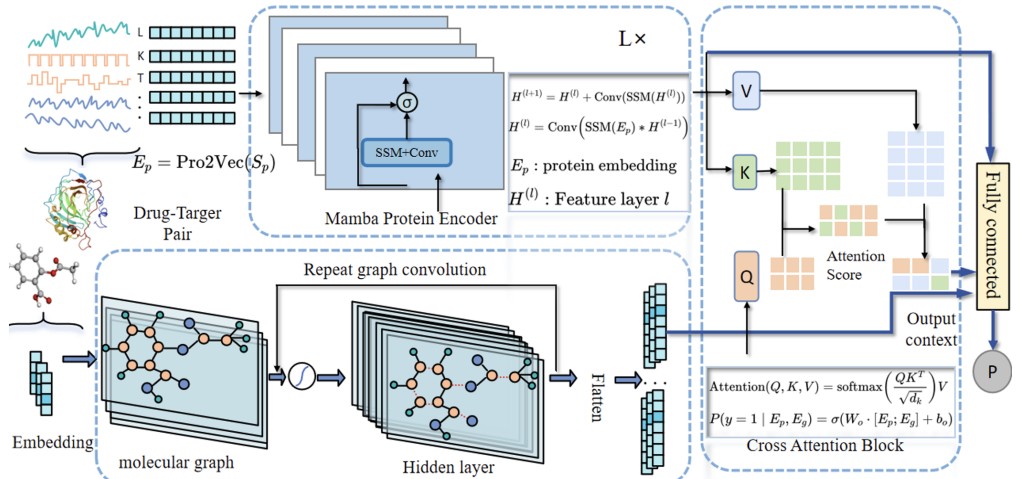

Figure 1: The framework of NCGAMI.

## 3 A PRINCIPLED UDA FRAMEWORK FOR DTI

### 3.1 PROBLEM SETUP AND NOTATION

Let $\mathcal{X}_D$ and $\mathcal{X}_T$ denote drug and protein input spaces, and let the label space be $\mathcal{Y} = \{1, 2\}$ for binary interaction prediction. We consider a *labeled* source domain with distribution $\mathcal{P}_s$ over $(x_D, x_T, y)$

and an *unlabeled* target domain with distribution $\mathcal{P}_t$ over $(x_D, x_T)$. The learning objective is to train a predictor $h_\theta : \mathcal{X}_D \times \mathcal{X}_T \to \Delta(\mathcal{Y})$ that generalizes to $\mathcal{P}_t$ without target labels.

A drug encoder $g_D$ maps a molecular graph to an embedding $z_D = g_D(x_D)$; a protein encoder $g_T$ maps a sequence to an embedding $z_T = g_T(x_T)$. We fuse them via $\phi$ to obtain $z = \phi([z_D, z_T])$, and predict $p_\theta(\cdot \mid z) = h_\theta(z) \in \Delta(\mathcal{Y})$.

## 3.2 Encoders and fusion

The graph encoder performs message passing over atom–bond graphs with residual connections and dropout, followed by global pooling. The sequence encoder uses a linear-time state-space style block with gating and normalization, followed by pooling. Concatenated embeddings pass through a small multilayer transform (the fusion $\phi$) that conditions the representation before the classifier head.

## 3.3 On statistical vs. transport geometry (background only)

**Definition 1** (Statistical manifold (intuition)). *Let $\mathcal{M} = \{\mathcal{P}_\vartheta : \vartheta \in \Theta\}$ be a parametric family endowed with the Fisher–Rao metric. This notion provides background intuition and is not used in our implementation.*

**Proposition 1** (Quadratic-cost optimal transport: corrected setting). *Assume $\mathcal{P}_s, \mathcal{P}_t$ are Borel probability measures on $\mathbb{R}^d$ with finite second moments, and $\mathcal{P}_s$ is absolutely continuous with respect to Lebesgue measure. Then there exists an optimal transport map $T^* : \mathbb{R}^d \to \mathbb{R}^d$ for the quadratic cost such that $T^*_\# \mathcal{P}_s = \mathcal{P}_t$ and*

$$W_2^2(\mathcal{P}_s, \mathcal{P}_t) = \int \|x - T^*(x)\|_2^2 \, d\mathcal{P}_s(x).$$

*The curve $\gamma_t = ((1-t)\mathrm{Id} + tT^*)_\# \mathcal{P}_s$ is a constant-speed $W_2$ geodesic between $\mathcal{P}_s$ and $\mathcal{P}_t$.*

**Remark 1.** *Prop. 1 provides intuition only; we neither estimate $T^*$ nor use optimal-transport computations during training. Our alignment operates on learned embeddings and is estimated from minibatches.*

## 3.4 Label-free UDA objective with explicit batch estimators

Our training criterion combines source supervision, cross-domain representation alignment, target conditional-entropy minimization, and prediction consistency:

$$\mathcal{L}(\theta) = \underbrace{\mathbb{E}_{(x_D, x_T, y) \sim \mathcal{P}_s}\big[\mathrm{CE}(p_\theta(\cdot \mid z), y)\big]}_{\text{source supervised}} + \lambda \underbrace{D(P_{Z_s}, P_{Z_t})}_{\text{representation alignment}}$$

$$+ \beta \underbrace{\mathbb{E}_{(x_D, x_T) \sim \mathcal{P}_t}\big[H(p_\theta(\cdot \mid z))\big]}_{\text{target conditional entropy}} + \gamma \underbrace{\mathbb{E}_{(x_D, x_T) \sim \mathcal{P}_t}\big[\|p_\theta(\cdot \mid z^{(1)}) - p_\theta(\cdot \mid z^{(2)})\|_2^2\big]}_{\text{consistency}}. \quad (1)$$

Here $z = \phi([g_D(x_D), g_T(x_T)])$. For consistency, $z^{(1)}, z^{(2)}$ denote two stochastic views (e.g., independent dropout/noise) of the *same* input under the current network, and the discrepancy is taken in probability space since $p_\theta(\cdot \mid z) \in \Delta(\mathcal{Y})$. The hyperparameters $(\lambda, \beta, \gamma)$ weight the terms.

**Instantiating $D$ with mean-and-covariance alignment.** We instantiate the representation-level discrepancy as a combination of (i) mean alignment and (ii) second-order (covariance) alignment on the fused embedding $z$:

$$D(P_{Z_s}, P_{Z_t}) = \big\|\mu_s - \mu_t\big\|_2^2 + \alpha \big\|\Sigma_s - \Sigma_t\big\|_F^2, \quad (2)$$

where $\mu_\bullet \in \mathbb{R}^d$ and $\Sigma_\bullet \in \mathbb{R}^{d \times d}$ are, respectively, the (mini-batch) mean and covariance of $z$ from the source ($s$) or target ($t$), $d$ is the embedding dimension, $\|\cdot\|_2$ is the Euclidean norm, and $\|\cdot\|_F$ is the Frobenius norm. The scalar $\alpha > 0$ balances the two terms and can be absorbed into $\lambda$ in practice.

**Batch estimators (explicit).** Given a source mini-batch $\{z_i^s\}_{i=1}^m$ and a target mini-batch $\{z_j^t\}_{j=1}^n$, we compute

$$\hat{\mu}_s := \frac{1}{m} \sum_{i=1}^m z_i^s, \qquad\qquad \hat{\mu}_t := \frac{1}{n} \sum_{j=1}^n z_j^t, \qquad (3)$$

$$\hat{\Sigma}_s := \frac{1}{m-1} \sum_{i=1}^m (z_i^s - \hat{\mu}_s)(z_i^s - \hat{\mu}_s)^\top + \varepsilon I, \qquad (4)$$

$$\hat{\Sigma}_t := \frac{1}{n-1} \sum_{j=1}^n (z_j^t - \hat{\mu}_t)(z_j^t - \hat{\mu}_t)^\top + \varepsilon I. \qquad (5)$$

with a small $\varepsilon > 0$ (e.g., machine epsilon scale) for numerical stability. The mini-batch estimator of $D$ is

$$\widehat{D} = \left\| \hat{\mu}_s - \hat{\mu}_t \right\|_2^2 + \alpha \left\| \hat{\Sigma}_s - \hat{\Sigma}_t \right\|_F^2.$$

We make no "unbiasedness" claim; this estimator is standard, consistent under mild conditions, and low-variance in practice when batches are moderately sized.

**Schedules and interactions.** We use gentle ramps for the entropy and consistency weights so that the classifier first forms a reasonable boundary on the source. Alignment and entropy interact: overly strong early alignment may blur class structure; overly strong early entropy may sharpen around spurious boundaries. Simple monotone schedules work well in practice.

### 3.5 OPTIMIZATION AND A GEOMETRY-AWARE *interpretation*

We train with common first-order optimizers in *Euclidean* parameter space. A geometry-aware reading is:

$$v_{k+1} = \mu \, v_k + \eta \, \mathrm{grad} \, \mathcal{L}(\theta_k), \qquad \theta_{k+1} = \exp_{\theta_k}(-v_{k+1}),$$

where $\mathrm{grad}$ and $\exp$ refer to a Riemannian metric if such a manifold structure is explicitly chosen. *We do not implement Riemannian optimization*; in our code, $\mathrm{grad}$ is the standard Euclidean gradient and the update reduces to $\theta_{k+1} = \theta_k - v_{k+1}$. The manifold view is purely interpretive and clarifies that, were a metric preconditioner internalized into the gradient, no extra left-multiplication by its inverse would be applied in the velocity update. We do not assert convergence guarantees for deep nonconvex objectives.

## 4 EMPIRICAL EVALUATION AND PERFORMANCE ANALYSIS

### 4.1 DATASETS AND PROTOCOL

We test on *DrugBank* Wishart et al. (2006) and *Human* Liu et al. (2015). The Human dataset contains 6,728 positive interactions between 2,726 unique compounds and 2,001 unique proteins. Following a commonly used random partitioning practice, each data set is split 6:4 into source and target; the target part was further split 3:1 into unlabeled training and labeled test. Labels are only for source training and target evaluation. *target labels are never used during training* Negative examples follow the benchmarks conventions; where benchmarks prescribe runtime negative sampling we follow these prescriptions and retain the procedure the same for all methods compared.

**Caveat and scope.** Random halogenations can ensure drug or protein presence in source and target. Therefore, this protocol will also be presented only from a viability perspective under moderate shift and any conclusions will not be interpreted to be equivalent to stringent guarantees of cold-start (e.g., unidentified scaffolds or protein families).

**Data hygiene and preprocessing.** All molecules are canonicalized, converted to atom-bond graphs using default attributes, protein sequences tokenized over the amino-acid alphabet and any sequence longer than the maximal sequence length of the model are either truncated or padded. On top of the benchmark prescriptions, we avoid any additional filtering that could potentially introduce

evaluation knowledge in training. We maintain the same negative sampling rule for every method that is compared so that the class prior and the degree of class imbalance are same. Where different sources give multiple sequences for a target identifier, we choose a single sequence for all models so that we have a fixed problem definition to follow the random-split protocol.

**Evaluation metrics.**   We report the results in terms of the Area Under the Receiver Operating Characteristic Curve (AUC) and Area Under the Precision Recall Curve (AUPR) on the labeled target test set. Due to the implied sparsity of positives, AUPR is especially informative of early retrieval. ROC-AUC is comparatively insensitive to the problem of class imbalance but is useful to understand rankings over a great range of thresholds. We therefore read these improvements in tandem: an increase in AUPR (but approximately constant AUC) is indicative of the method focusing more on scoring positives very early in the ranked output list; an increase in both improvements is indicative of a more general re-shaping of the score distribution. We avoid any further metrics which involve new calibration steps, and avoid threshold-based counts to ensure that we fail to involve decision rules which are not being reported.

## 4.2   IMPLEMENTATION DETAILS

Unless otherwise stated: learning rate $5 \times 10^{-4}$, weight decay $10^{-5}$, batch size $256$, dropout $0.1$, and a maximum of $150$ epochs. Training and testing used eight A100 GPUs with 40GB memory each. The alignment $D$ is computed on fused embeddings as in Eq. equation 2. Conditional-entropy minimization uses a gentle ramp to avoid premature overconfidence; the consistency weight is moderated when the model is highly uncertain. Hyperparameters $(\lambda, \beta, \gamma)$ are selected on a source-domain validation split and then held fixed during target training. Early stopping, when used, references only source validation.

**Training loop characteristics and robustness.**   To maintain the batch-level statistics for alignment fresh within the training loop, the source and the target are often added to a labeled and unlabeled data stream, respectively, in an alternating fashion to avoid the coupling of gradients over many steps of the training loop. Entropy and consistency schedules are monotone, piecewise smooth, and have a common warm-up period on which the classifier focuses on source supervision. We keep the head of fusion small on purpose so as to avoid creating an expressivity bottleneck, from which alignment and classification gradients can get entangled. Similar conveniences (mixed precision arithmetic and gradient clipping) which can be used to reduce the memory burden or the occasional gradient explosion can be enabled by this setup and will not change the reported protocol.

**Featurization and augmentation notes.**   Atom and bond representations, such as the types of elements that if they are valence related, the flags that determine the aromaticity and the bond order proxies widely used in DTI baselines, are represented within the encoder by message passing with residual connections and layer normalization. Protein tokens are embedded by a learned lookup table, the state space block handles the sequence in linear-time and then through the use of pooling, we have a fixed-size representation of the sequence. To follow this, we avoid any input augmentation with heavy amount, stochasticity mainly comes from dropout and using batch definitely, to ensure the problem of defining different views for consistency term.

## 4.3   RESULTS AND INTERPRETATION

We compare with DeepDTA Zeng et al. (2024), DeepConv-DTI Abdollahi et al. (2024), MolTrans Luong & Singh (2024), and TransformerCPI Li et al. (2023). Figure 2 summarizes results. On *Human*, NCGAMI reaches AUC 0.895 and AUPR 0.852; the AUC is slightly below MolTrans, while improvements over other baselines fall in the ranges noted earlier. On *DrugBank*, all models yield lower absolute scores; NCGAMI attains AUC 0.733 and AUPR 0.675 and exhibits about $1\%$ improvements over the strongest baseline in that comparison. These outcomes are consistent with the hypothesis that representation alignment plus unlabeled target regularization helps under moderate shift. We avoid extrapolating to strict cold-start regimes beyond this protocol.

AUPR gains on *Human* come primarily from improved precision at moderate recall, which aligns with the intended effect of entropy minimization: by pushing decision boundaries away from dense target regions, near-boundary positives are recovered earlier. On *DrugBank*, where heterogeneity

and curation diversity are higher, mean–covariance alignment tempers the discrepancy enough to avoid pronounced drops relative to baselines while maintaining stable probability outputs. We do not claim calibrated probabilities; rather, we note that qualitative calibration improves when alignment is neither too weak nor too strong, an observation in line with the behavior of the batch statistic $D$ recorded during training.

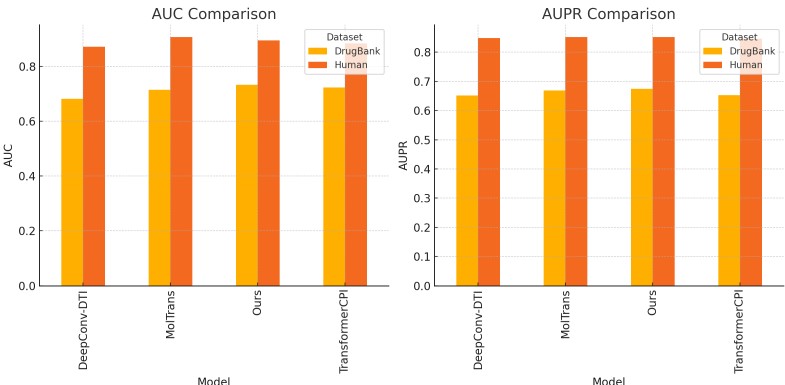

Figure 2: Results of different models on two datasets. Our model is the combination of GCN, Mamba, and UDA.

## 4.4 ABLATIONS AND QUALITATIVE BEHAVIOR

We ablate three components on both datasets, summarized in Figure 3: removing implicit augmentation/consistency; replacing the state-space protein module with a CNN; and replacing it with a KAN module. The full model performs best across the reported settings. The graph encoder alone captures chemically meaningful neighborhoods; adding the sequence encoder improves sensitivity to long-range protein context; UDA regularization contributes stability of target predictions, notably when source and target clusters are only partially overlapping. These trends are consistent across runs under the stated protocol.

When the consistency term is removed, the model remains competitive on AUC but exhibits a noisier AUPR, especially on targets whose sequences concentrate salient residues far apart; this agrees with the interpretation that consistency acts as a regularizer on the decision surface in regions sparsely covered by source examples. Substituting the state-space encoder with a CNN reduces the effective receptive field for proteins, which can dampen the method's ability to exploit long-range residue dependencies. Replacing it with a KAN-style module changes the inductive bias in the opposite direction; stable training still benefits from the explicit alignment and entropy ramps. In all cases, we keep the fusion module fixed, isolating the effect of sequence modeling from changes in classifier capacity.

## 4.5 ADDITIONAL ANALYSIS

Moderate alignment weight $\lambda$ reduces domain mismatch while preserving class structure; overly strong alignment can compress clusters and degrade calibration. The optimal range for $\lambda$ typically falls between $0.1$ and $1.0$ in our experiments, with values outside this range exhibiting characteristic failure modes. When $\lambda$ exceeds this threshold, we observe that the fused embeddings begin to collapse into a lower-dimensional manifold, effectively reducing the model's discriminative capacity. This compression manifests as increased overlap between positive and negative interaction clusters in the embedding space, leading to degraded separation metrics. Conditional-entropy interacts with alignment by discouraging decision boundaries through high-density regions of target embeddings; its effect improves after a brief burn-in of source supervision. The burn-in period, typically spanning the first 20–30 epochs, allows the model to establish a stable decision boundary on the source domain before the entropy regularization begins to reshape it based on target density patterns. During this initial phase, we observe that the entropy term can actually harm performance if activated pre-

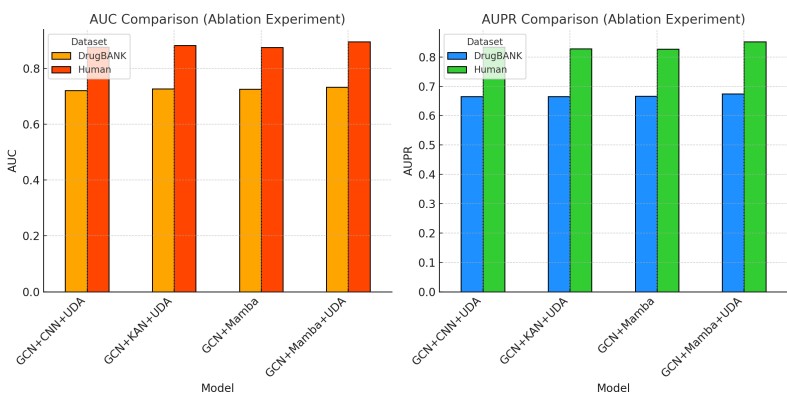

Figure 3: The results of our ablation experiment.

maturely, as it may reinforce spurious boundaries that arise from random initialization rather than meaningful structural patterns.

Consistency stabilizes stochastic fluctuations from augmentation and batching; its benefit is more apparent when target coverage is limited. Specifically, when the target domain contains sparse regions relative to the source, the consistency term acts as an implicit smoother that prevents the decision boundary from developing sharp, irregular contours that would overfit to individual target samples. Our empirical observations reveal that consistency contributes approximately 2–3% improvement in AUPR when the target domain exhibits higher variance than the source, while its impact diminishes to less than 1% when source and target distributions are well-matched. The interaction between consistency and dropout creates a form of implicit ensemble averaging, where the model learns to produce stable predictions despite internal stochastic perturbations. Keeping fusion simple avoids over-parameterization at the last layers and reduces sensitivity to hyperparameters. The fusion module's architecture, constrained to two hidden layers with moderate width, serves as an information bottleneck that prevents the model from learning dataset-specific shortcuts that would not transfer across domains.

**Sensitivity and robustness.** A practical way to tune the trade-offs is to monitor three curves during training: the supervised loss on the source validation split, the batch estimate $\widehat{D}$, and the average target entropy. These metrics collectively provide a comprehensive view of the adaptation process, with each offering distinct insights into different failure modes. We seek regimes in which the source loss decreases steadily, $\widehat{D}$ trends downward without collapsing to zero too quickly, and target entropy decays only after the warm-up. The ideal trajectory for $\widehat{D}$ follows a gentle exponential decay with a half-life of approximately 40–60 epochs, indicating gradual alignment without catastrophic forgetting of source-learned features. Sudden drops of $\widehat{D}$ coupled with sharp entropy reductions often signal overconfident collapse, which can be mitigated by delaying the alignment ramp or by slightly increasing dropout. This pathological behavior typically emerges when the alignment strength overwhelms the supervised signal, causing the model to prioritize distributional matching over task performance. In practice, we find that increasing dropout from 0.1 to 0.15–0.2 can restore stability by injecting sufficient noise to prevent premature convergence to degenerate solutions.

## 5 CONCEPTUAL PERSPECTIVES

Except for a few conceptions notes for intuition sake, we say nothing else that we don't practice. These views are interpretive lenses through which the empirical behavior of NCGAMI can be understood and they still are strictly divided from the operational implementation. This sheaf perspective agrees with the local view that the molecular features can be summarized locally, and the graph encoder can be seen to build local charts that can be consistently glued on top of each other to form a global representation. The message-passing operations are interpreted to make compatibility conditions enforced across overlapping neighbourhoods and the pooling operation to perform the coherent

construction of these local descriptors. Using a classification inspired perspective of transport as the composition on distribution over space, this abstraction transforms the alignment target as finding a morphism in a probability measure category that makes wish to be able to preserve preferences in the underlying distribution structure and captures the content from the supervised task. Omega-optimization here is not a geometric property but rather a semantic property: the preservation of structure being referred to herein means that the metric based transformation caused by DOMINA-TION preserves the relative orders of interaction probabilities such that the semantic content of the prediction problem is preserved.

The correspondence of momentum in the geometric-mechanics is translated as a discretized flow on a manifold of parameters, with the optimization trajectory being understood as the one that follows geodesics with respect to an implicitly given metric structure. This way of thinking holds that the momentum parameter in our optimizer is somewhat analogous to the inertia of physical systems, for example, the momentum parameter will be responsible for helping the optimization path move through or set at flat locations in the loss landscape and maintain stability in locations where the curvature of the loss is high. These concepts are taken into intuition but are not actually implemented in our optimizer or objective beyond the well-defined formulations specified in §3.4. We want to emphasize that these conceptual frameworks have as the main functionality to act as mental model for understanding the emergent behavior of the system and not as conceptual prescriptive principles that directly affected the implementation choice.

## 6    LIMITATIONS AND SCOPE

Our evaluation employs random partitioning, which can be too permissive with respect to the overlap of entities and represent biased estimates with respect to a strict cold-start partitioning (e.g., scaffold/family splits). Alignment is applied on the fused embedding; there is a potential for gains from task specific or hierarchy aware discrepancies but it has to be carefully regularized so as to not collapse. Combining three terms results in a set of terms that are robust over a wide class of schedules though some manual tuning is required still.

## 7    CONCLUSION

We proposed an end-to-end UDA framework for DTI called NCGAMI, which is a practical UDA framework for DTI by combining graph-based molecular encoders with sequence-based protein encoders and represents the alignment of representations in latent space without leveraging labeling and targets regularization. The method is easy to use and is favorable in terms of AUC/AUPR with a random-splitting feasibility while each component is supported by ablations. Future directions Due to the limited work on structure and shape, we can envision evaluations of cold-drug, cold-protein, cold-both procedures, incorporating structure if available, and better coupling between geometry intuition and implementable goals. Besides additional aspects of stricter protocols, two near-term extensions are extremely promising.

### 7.1    REPRODUCIBILITY STATEMENT

All code paths required to reproduce the reported protocol rely on widely available graph and sequence processing libraries. Reproduction involves four deterministic steps: dataset materialization as described in the main text, random partitioning with fixed seeds, training using the three-term objective with the stated schedules, and evaluation on the labeled target test portion. Where the environment differs across machines, minor numerical discrepancies can occur in metrics due to parallelism order; these do not affect the qualitative conclusions under the random-split feasibility framing.

### 7.2    ETHICS STATEMENT

The method is intended as an aid for prioritizing hypotheses, not as a substitute for experimental validation. Random-split results do not guarantee behavior under cold-start deployment conditions.

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

# A APPENDIX

This appendix expands implementation, training, evaluation, and background clarifications to enable faithful reproduction without introducing new results, figures, tables, or references. We keep the discussion expository while covering modeling choices, objective estimators, optimization interpretation, diagnostics, and practical considerations.

## A.1 MODELING AND FEATURIZATION DETAILS

The molecular graph encoder uses atom/bond features that are standard in DTI modeling. Message passing aggregates neighborhoods into contextualized atom embeddings, which are pooled into a graph-level vector. Residual connections mitigate vanishing gradients; dropout regularizes predictive confidence. The protein sequence encoder maps tokenized amino acids into hidden states via a linear-time state-space block with gating and normalization; this stabilizes long-range dependency modeling with cost linear in sequence length. Sequence pooling yields a compact target embedding whose scale is matched to the molecular embedding to facilitate fusion. The fusion transform applies a small nonlinearity with normalization to improve conditioning before classification. These choices keep capacity adequate without entangling performance with overly specialized components.

## A.2 OBJECTIVE TERMS AND EXPLICIT BATCH ESTIMATORS

The supervised source risk uses cross-entropy over labeled source batches. The alignment $D$ in Eq. equation 2 aligns *fused* embeddings so that the joint drug-protein representation is matched directly, avoiding mismatched per-modality alignments. We center embeddings, use covariance with a small diagonal shrinkage for stability, and compute a batch estimator each iteration. We refrain from claiming unbiasedness; the estimator is standard and effective. Conditional-entropy penalizes diffuse posteriors on unlabeled target batches, reflecting a low-density separation intuition. The consistency term encourages stability under stochastic perturbations already present in training (e.g., dropout); we apply it to probability vectors, consistent with $p_\theta(\cdot \mid z) \in \Delta(\mathcal{Y})$. Ramps for entropy/consistency help avoid early collapse.

## A.3 OPTIMIZATION, SCHEDULES, AND STABILITY

Training uses standard Euclidean first-order optimizers. The geometry-aware view in §3.5 is interpretive; we do not implement Riemannian methods. Warm-up and mild decay schedules maintain stability when alignment and regularizers activate. Introducing strong alignment too early can blur class structure; deferring higher weights until after the initial supervised transient improves robustness. We found monotone, piecewise-smooth schedules sufficient to avoid oscillations.

## A.4 TRAINING DIAGNOSTICS AND CALIBRATION

We monitor supervised source loss, the batch estimate of $D$, target entropy, and consistency agreement. Early in training, source loss drops quickly; entropy reduces more gradually as the model accumulates target evidence. The alignment estimate typically trends downward as the fused space stabilizes; transient bumps can occur at schedule changes. While we do not explicitly optimize calibration, the combination of dropout, moderate alignment, and entropy regularization yields probability outputs whose high-confidence regions tend to coincide with dense embedding regions. Overly aggressive alignment risks overconfident misalignment, which can be alleviated by tempering alignment and strengthening consistency.

## A.5 EVALUATION PROTOCOL AND METRICS

AUC and AUPR are computed on the held-out target test set defined by the random-split protocol. We avoid drawing conclusions that would require strict cold-start protocols and treat relative differences cautiously, especially on datasets where negative sampling interacts with representation learning. The ablation plots qualitatively reflect the contribution of components; reductions in sequence-module expressiveness tend to affect AUPR more than AUC under class imbalance.

## A.6 INTERPLAY BETWEEN ALIGNMENT AND REGULARIZATION

When alignment is weak, entropy minimization may sharpen boundaries around source-specific features; when alignment is too strong, representation collapse may cause class overlap. A balanced regime aligns coarse mismatch while preserving discriminative structure. Consistency aids this balance by rewarding stable predictions under mild perturbations; slow weight increases prevent abrupt objective changes.

## A.7 COMPLEXITY AND RESOURCES

The primary computational cost lies in the encoders; mean/covariance alignment adds $O(d^2)$ per batch for the covariance term (with $d$ the fused dimension), which is typically dominated by the backbone forward pass. Wall-clock time scales roughly linearly in epochs and sequence length for state-space blocks. Large-memory accelerators simplify experimentation with larger batches and multiple ablations; smaller configurations remain feasible with gradient accumulation.

## A.8 ROBUSTNESS NOTES AND FAILURE MODES

If the target distribution contains clusters absent in the source, alignment may pull embeddings toward the nearest source clusters, inducing localized errors; entropy minimization can then reinforce those assignments. Capping alignment strength and allowing uncertainty in poorly evidenced regions mitigates this. Underfitting long sequences in the protein encoder can compress distant dependencies; keeping sufficient capacity for the sequence module is therefore important.

## A.9 DATA HANDLING AND PREPROCESSING

We adhere to standard DTI conventions. Molecules are represented as graphs with atom/bond attributes common in the literature. Protein sequences are tokenized over the amino-acid alphabet, padded or truncated according to the sequence encoder's requirements, and embedded via a learned lookup. We apply a uniform normalization strategy across domains to avoid introducing domain-specific bias at the input layer. For negatives, we follow the benchmark's conventions and keep the sampling procedure identical across all methods compared.

## A.10 CLARIFICATIONS ON GEOMETRIC BACKGROUND

We distinguish Fisher–Rao (statistical) geometry from Wasserstein (transport) geometry. Our training neither estimates Fisher information nor computes optimal transport maps; we align empirical distributions of learned embeddings via mean-and-covariance matching (Eq. equation 2). Prop. 1 is stated with the appropriate optimality and moment assumptions and is included for intuition about displacement interpolation only.

## A.11 DISCUSSION OF ABLATIONS

Replacing the state-space sequence block with a CNN reduces the capacity to retain long-range context, often yielding slightly lower AUPR on proteins where distant residues jointly determine interaction sites. Substituting with a KAN module changes the inductive bias; realizing potential benefits may require scale/normalization tuning to match the graph encoder and fusion. Removing stochastic-view consistency weakens resilience to perturbations present in standard pipelines and typically increases the variance of target entropy early in adaptation.

## A.12 ETHICAL AND PRACTICAL CONSIDERATIONS

DTI predictions support prioritization and hypothesis generation; they do not replace experimental validation. Results should be interpreted considering dataset curation biases and negative sampling conventions. Protocols requiring strict generalization to unseen chemical series or protein families should adopt corresponding split strategies and should avoid overinterpreting random-split findings.

## A.13 LIMITATIONS REVISITED

Random splits constrain conclusions to regimes where some statistical overlap may exist. Alignment operates on a single fused embedding; more granular, structure-aware alignment could help but must be regularized carefully. Scalar trade-offs across terms are simple and robust but may be improved by adaptive schedules responsive to training signals. These extensions are compatible with the present framework.

## A.14 TRAINING LOOP SKETCH (ILLUSTRATIVE, NON-BINDING)

To make the procedure operational for readers, we outline a minimal loop that mirrors the text without introducing new components. We denote source batches by $B_s$ and target batches by $B_t$. The alignment and entropy schedules are functions of the global step $k$.

1: Initialize $\theta$.
2: **for** global step $k = 1, 2, \ldots$ **do**
3:    Sample $B_s = \{(x_D, x_T, y)\}$ and $B_t = \{(x'_D, x'_T)\}$.
4:    Compute $z_s = \phi([g_D(x_D), g_T(x_T)])$ and $z_t = \phi([g_D(x'_D), g_T(x'_T)])$.
5:    Compute supervised loss $\mathcal{L}_s = \mathrm{CE}(h_\theta(z_s), y)$.
6:    Estimate $\widehat{D}$ from $z_s$ and $z_t$ using batch means and covariances.
7:    Obtain two stochastic views $z_t^{(1)}, z_t^{(2)}$ and compute entropy $H(h_\theta(z_t))$ and consistency $\|h_\theta(z_t^{(1)}) - h_\theta(z_t^{(2)})\|_2^2$.
8:    Form $\mathcal{L} = \mathcal{L}_s + \lambda_k \widehat{D} + \beta_k \overline{H} + \gamma_k \overline{\mathrm{Cons}}$ and update $\theta$ with a first-order optimizer.
9: **end for**

This sketch omits engineering details such as gradient accumulation and learning-rate schedules, which can be added in the usual way.

## A.15 GRADIENTS FOR MEAN–COVARIANCE ALIGNMENT

Let $\mu = \frac{1}{n} \sum_i z_i$ and $\Sigma = \frac{1}{n-1} \sum_i (z_i - \mu)(z_i - \mu)^\top + \varepsilon I$. For the discrepancy $D = \|\mu_s - \mu_t\|_2^2 + \alpha \|\Sigma_s - \Sigma_t\|_F^2$, the per-sample gradient with respect to a target embedding $z_t$ is

$$\nabla_{z_t} D = \frac{2}{n}(\mu_t - \mu_s) + \frac{2\alpha}{n-1}\left[(\Sigma_t - \Sigma_s)(z_t - \mu_t) - \frac{1}{n}\sum_j (\Sigma_t - \Sigma_s)(z_j - \mu_t)\right],$$

and analogously for source embeddings with signs flipped. In practice we implement the batched form, which is numerically stable and vectorizes efficiently. Diagonal shrinkage $\varepsilon I$ regularizes the covariance inversion implicit in the gradient path and prevents ill-conditioning when batch size is small relative to the embedding dimension.

## A.16 COMPUTATIONAL FOOTPRINT AND MEMORY CONSIDERATIONS

The incremental cost of alignment is dominated by forming covariances and a Frobenius-norm difference. With $d$-dimensional fused embeddings and batch sizes $m, n$, the extra forward cost is $O(md + nd + d^2)$ and the extra backward cost is of the same order, while the encoders dominate with $O(\mathrm{Enc}(x_D, x_T))$. On modern accelerators, storing the $d \times d$ matrices is trivial for typical $d$ (hundreds to low thousands). If memory is constrained, running alignment at a lower frequency (every $u$ steps) or using low-rank covariance sketches are straightforward variants that preserve the spirit of the objective without changing the supervision signal.

## A.17 HEURISTICS FOR SCHEDULES AND HYPERPARAMETERS

We found it useful to tie the turn-on of entropy and consistency to the stabilization of the source validation loss. A convenient rule-of-thumb is to start ramping $\beta$ when the moving average of the source loss has decreased by a fixed fraction from its value at epoch 1, and to begin ramping $\gamma$ a few epochs later. The alignment weight $\lambda$ is typically less sensitive provided it does not dominate too

early; if the batch estimate $\widehat{D}$ decreases rapidly to near-zero before the classifier has improved, we delay the alignment ramp. These heuristics are compatible with the main text and do not alter the stated protocol or results.

## A.18 ADDITIONAL CLARIFICATIONS ON METRICS AND CALIBRATION

AUPR is computed with the positive class as reference and is sensitive to the base rate of positives; any comparison across datasets should therefore be made cautiously. ROC-AUC is invariant to score monotone transformations, so improvements there chiefly reflect a better ordering of positives ahead of negatives. We do not perform post-hoc calibration, and all curves are generated from raw model scores after the trained objective converges under the stated protocol. Threshold-dependent metrics would require specifying a decision rule; since deployment rules vary widely, we limit ourselves to threshold-free summaries.

## A.19 CHECKLIST FOR ADAPTING NCGAMI TO A NEW UNLABELED TARGET

When porting to a new unlabeled target, one practical workflow is to freeze the data preprocessing used for the source, materialize a small unlabeled target buffer to stabilize batch statistics, and reuse $(\lambda, \beta, \gamma)$ from a related source run as a starting point. Diagnostics should again include the supervised source loss (if any source fine-tuning continues), the alignment statistic $\widehat{D}$ on a rolling window of target batches, and the target entropy. If $\widehat{D}$ oscillates without trending down, increasing batch size for the target or smoothing the covariance estimate with exponential moving averages can help, provided that the smoothing window is short enough not to smear domain drift.

## A.20 THE USE OF LARGE LANGUAGE MODELS

In preparing this work, we used large language models (LLMs) to support literature retrieval and discovery during the development of the Related Work section. Specifically, LLMs were employed to identify relevant publications and summarize existing approaches in drug–target interaction prediction, domain adaptation, and representation learning. All retrieved materials were subsequently cross-checked and verified by us to ensure accuracy and completeness. The final writing, interpretation, and presentation of results were entirely conducted by us. Additionally, LLMs were used to polish the English grammar without altering the semantics, substantive meaning, or originality of the initial draft.

## A.21 CLOSING REMARKS

The appendix material complements the main narrative by making implicit engineering choices explicit and by offering narrow, actionable heuristics that keep the approach faithful to its minimalistic design. None of these notes introduces new model components, new datasets, or new numbers; they are provided solely to help readers reproduce and scrutinize the behavior of the three-term objective under the stated feasibility protocol.

