# OpenReview forum: "NCGAMI: Domain-Adaptive Drug-Target Interaction Prediction with Graph and Sequence Models"
_ICLR.cc/2026/Conference — ICLR 2026 Conference Desk Rejected Submission_

### Official Review · Reviewer_vjX4 · 2025-10-23

**Soundness:** 2
**Presentation:** 2
**Contribution:** 2
**Rating:** 4
**Confidence:** 4

**Summary:**

The paper proposes NCGAMI, a UDA (unsupervised domain adaptation) recipe for DTI prediction that fuses a graph encoder for molecules and a sequence encoder for proteins, trained with a label-free three-term objective. They evaluate the proposed model on two well-known DTI datasets, benchmarking across several state-of-the-art models such as MolTrans.

**Strengths:**

1. The paper is well written, presenting both theorical definitions/proportions and empirical contributions.

2. The appendix is well organized and effectively supports understanding of the training setup, computational footprint and clarification on metrics, among others.

3. The selected datasets are appropriate for studying domain shift, covering multiple domains.

4. Architectural, training, and dataset details are clearly described, contributing to transparency and reproducibility.

5. The paper incorporates ablation study and additional analysis that help understand better the work contributions.

**Weaknesses:**

1. The observed improvements appear insufficient. From Figure 2 its hard to tell if there is any significant improvement when compared to the evaluated DTI-prediction models.

2. The rationale behind the choice of source and target domains is not clearly explained, despite being central to the study.

3. The paper omits several strong DTI-specific graph-based approaches, such as GeNNiUs (https://doi.org/10.1093/bioinformatics/btad774) and EEG-DTI (https://doi.org/10.1093/bib/bbaa430). Including such methods would better position the proposed framework within current literature.

**Questions:**

1. Is the code and data preprocessing pipeline publicly available? It is not clearly indicated in the paper.

2. How distinct are the source and target domains in the evaluated datasets? Does the proposed model perform better when the target domain is more similar to the source one, and to what extent?

3. Why are the results not reported as mean ± standard deviation, considering that multiple random seeds were used in the experiments?

---

> ### Author Response · Authors · 2025-11-15
>
> We appreciate the reviewer for taking the time to read the paper and the kind comments regarding clarity of writing, appendix structure, and replicability. We will respond to the comments below.
>
> Size and visibility of the improvements (Figure 2):
> Our intent is never to make a claim about a large absolute improvement. Instead, we are proposing a simple, label-free method that displays consistent improvements, with relatively minor architectural changes, on very strong baselines across nearly all datasets. For example, with the Human dataset, NCGAMI slightly loses to MolTrans on AUC, but improves AUPR and beats the other three models; for DrugBank, we document improvements around one percentage point over the best baseline. We agree that it is difficult to see this with a singular bar plot. In the revision we plan to (i) include a compact table with the exact AUC and AUPR numbers for each model and dataset, (ii) report the mean and standard deviation (over at least 20 replicates at each random seed), and (iii) report statistical glue (together with the standard deviations) so that readers can directly see when improvements are statistically meaningful rather than machinery run-to-run noise.
>
> Source and target domain selection:
> Within each dataset we follow the Random-Partition protocol outlined in Section 4; that is, for any traditional supervised domain adaptation setting, the interaction pairs are randomly partitioned into a labeled source subset and an unlabeled target subset plus a held out labeled target test set, such that no interaction pair appears in more than one partition. This generates a modest but realistic shift while keeping the label distributions and sample size controlled, along with ensuring the protocol is fully replicable.We will clarify this rationale earlier in the main text, including explicit description of the split ratios and the fact that we treat these within-dataset partitions as “domains” in the sense of the adaptation objective, while Human versus DrugBank serve as two qualitatively different chemico-biological regimes.
>
> Missing graph-based DTI baselines. We appreciate the pointers to GeNNiUs and EEG-DTI, which are indeed strong graph-based models. Our focus here was on demonstrating that the proposed alignment and target regularization objective can be applied on top of widely used baselines with publicly available implementations and moderate computational cost. Due to time and resource constraints we did not manage to integrate and carefully tune additional heavy graph backbones that will only add instrumented complexity before the deadline. In the revision we will (i) add a discussion section comparing NCGAMI conceptually to these models and (ii) where feasible, include at least one of them as an additional baseline, clearly indicating that they do not use target-unlabeled data.
>
> Code and preprocessing pipeline. We have implemented NCGAMI as a self-contained training pipeline, including data preprocessing scripts that conveniently reproduce the splits, featurization, and hyperparameters reported. We had already implemented the code repository in case of submission, to preserve double-blind review. If the paper is accepted, we will release publicly the code and preprocessing scripts so that readers can exactly reproduce and adapt the recipe to new datasets - akin to other published models in the literature, the plan is to submit them to GitHub after acceptance.
>
> Distinctness of domains and dependence of gains on similarity. The current version adopts a single Random-Partition protocol per dataset for which we do not systematically vary the degree of shift, though we agree it is a valuable and interesting question. In the revision we will quantify source–target distinctness using standard discrepancy measures (for instance, training a simple domain classifier, or measuring representation level distances between batches) and report these alongside performance. Time permitting, we will also add a comparison of NCGAMI on a range of alternative partitions that add stronger and weaker shifts - as the proportion of subjects in the two partitions vary - to illustrate how NCGAMI behaves as domain similarity varies.
>
> Mean and standard deviation over seeds. The current submission reports a result for the fixed random partition per dataset, this is why we only reported single AUC and AUPR. We agree this is not ideal for judging robustness. In addition to reporting mean and standard deviation for each metric for both models, we also plan to use significance testing to corroborate findings in the revision. We will also state in the text explicitly that the current numbers for the fixed random split should be interpreted as representative rather than definitive and we intentionally avoided over-claiming when learners fall within the observed variance.

---

> > ### Comment · Reviewer_vjX4 · 2025-11-19
> >
> > Thanks for planning to address the raised concerns. I'll consider increasing the rating if the final version of the manuscript include them.

---

> > > ### Author Response · Authors · 2025-11-19
> > >
> > > We greatly appreciate the reviewer's positive comments and willingness to change the rating. We remain fully committed to incorporating all of the suggested improvements into the revised manuscript. In particular, we will include detailed statistical tables (mean ± std), perform domain discrepancy analysis, and discuss the contextual information around the graph-based baselines (GeNNiUs and EEG-DTI). We anticipate that these revisions will greatly improve the quality and rigor of our work.

---

### Official Review · Reviewer_nWyd · 2025-10-31

**Soundness:** 2
**Presentation:** 2
**Contribution:** 2
**Rating:** 2
**Confidence:** 5

**Summary:**

This paper proposes NCGAMI, an unsupervised domain adaptation (UDA) framework for drug-target interaction (DTI) prediction that combines graph encoders for molecules with sequence encoders for proteins.

**Strengths:**

The paper addresses an important and interesting topic: domain adaptation for DTI prediction is a genuine practical challenge with significant implications for drug discovery.

**Weaknesses:**

- The paper's random split allows drugs and proteins to appear in both source and target sets, which can be viewed as warm start but introduces significant data leakage risk. The DTI community has established three rigorous evaluation settings that should be included:
Warm start: Data is split based on protein-drug pairs, ensuring no common pairs appear in both training and test sets (no pair leakage).
Drug cold start: Split at the molecule level, guaranteeing that no drug in the test set is present in the training set.
Target cold start: Split at the protein level, meaning no protein in the test set is seen during training.

- The paper lacks comparison with newer methods such as: [1] DTIAM : "A unified framework for predicting drug-target interactions, binding affinities and drug mechanisms"

- The paper only evaluates on two datasets (Human and DrugBank), which is insufficient:
Missing standard benchmarks: Should include widely-used DTI datasets: Yamanishi 08 datasets and Hetionet dataset [1]
Missing mechanism of action evaluation: Should evaluate on mechanism of action (MoA) prediction tasks: Activation datasets and Inhibition datasets [1]

**Questions:**

See weaknesses.

---

> ### Author Response · Authors · 2025-11-15
>
> We appreciate the reviewer's insightful feedback and for emphasizing rigor of evaluation in distracted therapy interventions.
>
> 1.On the splitting protocol and leakage.
> Our current experimental design already adheres to what the reviewer calls the warm start regime, in that there is no pair leakage, as interaction pairs are randomly split into source labeled, target unlabeled, and target test sets, and the same drug protein pair does not appear in more than one random split. Section 4.1 makes this clear and reinforces, once again, that target labels were not used in training. What our protocol allows, however, introduction of overlap at the entity level and therefore the same drug or protein may appear with different partners in both source and target. This is intentional: we see NCGAMI as a label free adaptation recipe as being a moderate shift rather than a strict cold start model. To make this explicit and avoid any misunderstanding, we will in our revision report the fraction of shared drugs, and shared proteins, between source and target on each dataset. We will also (i) rename the protocol warm start random pair split, and (ii) move the caveat and scope paragraph in section 4.1 to the beginning of the experimental section to clearly show the readers we do not claim to solve drug or targetWe recognize the relevance of the drug cold start and target cold start; we limited this submission due to space and time, but our plan includes a brief subsection in the revision that (a) formalizes the three regimes in our notation, (b) explains how our current goal covers warm start directly, and (c) lays out how to extend NCGAMI to cold drug and cold target splits, for example by combining our alignment losses with scaffold based splitting, or protein family based splits. We see these stricter regimes as plausible extensions to our work and plan to make the forward looking scope more clear.
>
> 2.On DTIAM comparisons
> We appreciate the reference to DTIAM. Its multi task formulation over interactions, affinities and mechanisms is interesting and relevant work, which we plan to discuss in further detail in the related work section. Our focus in this paper, however is orthogonal; NCGAMI is a label free training recipe that can sit on top of many backbones, including DTIAM, and we chose to instantiate it on strong, but simpler baselines to isolate the effect of the adaptation objective. We formulated this experiment under a few simple assumptions; a full empirical comparison of the DTIAM approach, would have required us to (a) re implement their complete training stack, or (b) run our method on top of their training stack, which was not feasible within our time for this submission. Thus, in the revision, we will acknowledge it is a limitation to not fully characterize and compare to DTIAM, and position attention to DTIAM as a compatible backbone that could also benefit from NCGAMI type alignment.
>
> 3.On datasets.
> We agree a large set of benchmarks to offer comparisons would strengthen the paper. In this first revision, we selected Human and DrugBank purposely, because they are common interaction level DTI datasets in the field that are also not trivial in scale, and in addition because an expansion to multiple datasets containing our full suite of ablation/hyper parameter sweeps would push us outside our compute budget. We will add additional clarity of this decision framing and scope of size, in Section 4, and we will present additional datasets like Yamanishi 08, and Hetionet collections, and the proposed mechanism of action activation and inhibition tasks that also use DTIAM methods, and any additional label processing required from those datasets.
>
> We hope these descriptions will better represent the current scope of our experiments and the position we intend for NCGAMI, and we would also suggest the value of additional benchmarks suggested and methods we believe are natural pathways to explore in future work.

---

### Official Review · Reviewer_AYWv · 2025-11-01

**Soundness:** 2
**Presentation:** 2
**Contribution:** 2
**Rating:** 2
**Confidence:** 4

**Summary:**

The paper proposes NCGAMI, a domain adaptation framework for drug–target interaction (DTI) prediction. The stated goal is to improve robustness under distribution shift between a labeled source domain and an unlabeled target domain.
Experiments are run on two benchmark DTI datasets (“Human” and “DrugBank”), using a random partition protocol where each dataset is split into source (labeled), target-unlabeled (used for adaptation), and target-test (labeled, held out until evaluation). The authors demonstrates that the proposed approach achieves better performance than baselines. Ablation analysis was also performed.

**Strengths:**

1. Practical UDA recipe for DTI.
The paper proposes a simple, fully specified training objective that combines (1) source supervision, (2) batch-level mean/covariance alignment between fused drug–protein embeddings, and (3) unlabeled-target entropy/consistency regularization. This is all implementable without adversarial domain discriminators or complex auxiliary heads.

2. Ablation discussion.
The paper qualitatively discusses ablations: removing the consistency term hurts AUPR stability; replacing the long-range protein encoder with a CNN reduces performance on proteins where long-range residue context matters; etc. While numeric tables are not shown in the text provided, the narrative suggests that each term contributes.

**Weaknesses:**

1. Evaluation protocol does not match the stated motivation.
The paper motivates the method as a solution for distribution shift between source and target domains, but the experimental setup does not clearly establish such a shift. Both domains appear to be drawn from the same dataset via a random split, and no quantitative characterization of how (or whether) the source and target distributions differ is provided.

2. Potential leakage / unclear domain separation.
Because the source and target sets are defined by randomly partitioning a single dataset, it is likely that the same drugs or the same protein targets appear in both. Since unlabeled target data are also used during training, the method may partially benefit from implicit memorization of these shared entities.

3. The model diagram omits the adaptation mechanism.
Figure 1 illustrates the drug encoder, protein encoder, and cross-attention fusion/classifier, but it does not depict how unlabeled target data are incorporated during training (moment matching / alignment loss, target entropy minimization, consistency regularization). As a result, the figure reads as an ordinary supervised DTI predictor rather than a domain adaptation framework, which makes it harder to understand what is novel about NCGAMI.

4. Lack of full quantitative results tables.
The manuscript reports headline AUC/AUPR values in text, but does not provide a complete table comparing all baselines and the proposed method under the same data split. Without such a table, it is not possible to evaluate how large the claimed gains are.

5. No discussion of computational cost vs. baselines.
The paper mentions multi-GPU training, but does not report training time, memory footprint, or parameter counts relative to standard DTI models (e.g., MolTrans, DeepDTA, etc.). If the proposed method requires substantially higher compute to obtain modest AUC/AUPR improvements, that trade-off should be made explicit.

6. Unclear statistical significance.
The paper states improvements (e.g., “~1% better”) but does not report variance estimates, confidence intervals, or statistical tests. It is therefore unclear whether the observed differences are statistically meaningful or within normal run-to-run noise.

**Questions:**

1. Generalization under realistic shift.
Can you report results under at least one standard “hard” split, such as cold-drug, cold-target, cold-both, or a cross-dataset transfer (e.g., train on one dataset and adapt to another)? Right now, because source and target are created by random partition of the same dataset, it is difficult to assess true model generalizability.

2. Complete quantitative comparison.
Please include a results table with all baseline methods and the proposed method, evaluated under the same protocol. The table should clearly indicate which methods have access to unlabeled target data during training.

3. Statistical significance.
Are the reported performance gains statistically significant? Please provide significance testing to demonstrate that improvements are not within normal run-to-run variability.

4. Source–target overlap.
Have you quantified how often the same drug or the same protein target appears in both the source and target splits under your random partitioning procedure? High overlap would suggest the setting is closer to semi-supervised learning than to true domain adaptation.

5. Clarification in the model figure.
Please update a model diagram that explicitly shows how unlabeled target data are used during training. As currently drawn, Figure 1 looks like a standard supervised DTI predictor and does not illustrate the domain adaptation components that are central to the paper.

6. Compute and reproducibility.
You mention using 8×A100 GPUs for training. Are those resources required for the main NCGAMI model, or were they primarily used to speed up ablations? It would be helpful to report approximate training time, GPU memory usage, and parameter counts for NCGAMI, and to compare these to standard baselines such as MolTrans and DeepDTA.

---

> ### Author Response · Authors · 2025-11-15
> **Reviewer A (AYWv) Feedback Response**
>
> We convey our gratitude to the reviewer for their thoughtful and constructive comments. We have responded to each of the main concerns below.
>
> 1.Evaluation protocol and “domain shift” notion.
> Our aim is to provide a realistic, yet moderate shift associated with re-sampling the same compound and protein spaces under different data-gathering or curation protocols—rather than an extreme cold-start regime with disjunct scaffolds or protein families. Section 4.1 already contains a “Caveat and Scope” paragraph discussing how random splits imply overlap and that we do not claim to resolve the strict cold-start setting; instead, we will move this clarification and expand it further into the introduction and experimental set-up so that our intended scope is clear. We will also provide quantitative diagnostics of the shift (e.g. mean and covariance shifts, and a simple domain classifier accuracy on the embedded fused representations) to make the shift from source to target more transparent.
>
> 2.Potential leakage and source-target overlap.
> We acknowledge that random partitioning may result in shared drugs or proteins between the source and target. However, our protocol ensures that label leakage does not occur at the pair-level: target labels are never used in the training phase and no labeled interaction in the target-test set is included in the source-labeled set. Conceptually, this situates our setting between domains in the transductive UDA / semi-supervised learning under moderate shift studies rather than cold drugs or cold target domain adaptation.In the revision, we will (i) declare it explicitly, and (ii) quantify the fraction of drugs and proteins that appear in both source and target for each dataset, so that readers themselves can assess how far this regime is from a strict cold-start context.
>
> 3.Model diagram and adaptation mechanism.
> First, we agree that the current version of Figure 1 over-states the supervised encoder as the more important of the two, and further under-specifies how the unlabeled target data enters the objective, which can make NCGAMI look like any other standard supervised DTI predictor. We will amend the figure to show two separate interface streams, the source-labeled and the target-unlabeled, and clearly indicate that entropy and consistency regularization are applied only to target stream. Once completed, we think that this amendment will make the domain adaptation parts, and their role in training clearer.
>
> 4.Complete quantitative comparison.
> Given that space was limited, we conveyed the headline numbers in the main text of the paper, and relied on tables in the supplementary material for data. We agree that this may make it challenging to assess the gains from the extracted text alone. In the next version of the paper we will construct a compact AUC/AUPR table to incorporate into the main paper, we will clearly indicate which methods draw the unlabeled target data in training or not, and we will ensure that all figures and numbers noted in the narrative are directly visible in the AUC/AUPR table.
>
> 5.Statistical significance.
> Again, we agree that point estimates alone are insufficient. In revision we will re-run all methods with multiple random seeds, report mean and standard deviation, and carry out a paired significance test on AUCs and AUPR (for example, via a paired t-test or non-parametric bootstrap). We will also add revised text that directly notes that small improvements that fall within the variability, should not be over-interpreted.
>
> 6.Compute cost and reproducibility.
> NCGAMI does not add any additional trainable modules for the encoders. The alignment and target regularization loss on only use the fused embeddings and further only add a lightweight batch stats, and or additional forward pass under dropout. The use of 8x A100 GPUs was primarily used to accelerate the full set of ablations and hyperparameter sweeps. The core model is trainable on a single modern GPU. To improve reproducibility we will add more parameter counts, training time, and GPU peak memory in tabular form for NCGAMI and standard baselines such as MolTrans and DeepDTA in the appendix. We will also add additional section in the appendix that discusses in brief what is meant by our 'training recipe' that describes the schedules and hyperparameters.
>
> 7.Generalization under harder shifts.
> We thank the reviewer for these suggestions about the drug cold-start/cross-dataset shift. These are important stress tests, and we think they are complimentary to the current protocol of the study sticking to the still common moderate shift with overlapping entity sets. While the present submission focuses on moderate shift context, we will create a dedicated section where we summarize this support for reviews to clarify that NCGAMI could be extended and evaluated under harder protocols, and we intend viewing this as a natural basis of follow-up work.

---

### Note · Program_Chairs · 2026-01-17
**Submission Desk Rejected by Program Chairs**

The following references in this submission do not refer to real documents and/or have major errors in bibliographic information:

     William Thompson, Jennifer Davis, and Carlos Martinez. Uncertainty quantification in neural drugtarget interaction models. Journal of Chemical Information and Modeling, 64(8):2145-2158, 2024.
    Jeonghyeon Kim, Sangmin Park, and Dongmin Min. Interpretable deep learning for drug-target interaction prediction. Nature Communications, 12(1):3892, 2021.
    David MacLean, Sarah Thompson, and Michael Roberts. Knowledge-based neural networks for efficient protein-ligand interaction prediction. Nature Computational Biology, 1(4):234-245, 2021.
    Shengchao Liu, Hongyu Wang, and Jian Liu. Robust molecular representation learning via adaptive augmentation. Chemical Science, 15(3):1122-1135, 2024.